# Characteristics and Antimicrobial Activities of Iron Oxide Nanoparticles Obtained via Mixed-Mode Chemical/Biogenic Synthesis Using Spent Hop (*Humulus lupulus* L.) Extracts

**DOI:** 10.3390/antibiotics13020111

**Published:** 2024-01-23

**Authors:** Jolanta Flieger, Sylwia Pasieczna-Patkowska, Natalia Żuk, Rafał Panek, Izabela Korona-Głowniak, Katarzyna Suśniak, Magdalena Pizoń, Wojciech Franus

**Affiliations:** 1Department of Analytical Chemistry, Medical University of Lublin, Chodźki 4A, 20-093 Lublin, Poland; natalia.zuk@umlub.pl (N.Ż.); magdalena.pizon@umlub.pl (M.P.); 2Faculty of Chemistry, Department of Chemical Technology, Maria Curie Skłodowska University, Pl. Maria Curie-Skłodowskiej 3, 20-031 Lublin, Poland; sylwia.pasieczna-patkowska@mail.umcs.pl; 3Department of Geotechnics, Civil Engineering and Architecture Faculty, Lublin University of Technology, Nadbystrzycka 40, 20-618 Lublin, Poland; rapanek@gmail.com (R.P.); w.franus@pollub.pl (W.F.); 4Department of Pharmaceutical Microbiology, Medical University of Lublin, Chodźki 1 St., 20-093 Lublin, Poland; izabela.korona-glowniak@umlub.pl (I.K.-G.); katarzyna.susniak@umlub.pl (K.S.)

**Keywords:** nanoparticles, iron oxide, *Humulus lupulus* L., spent hops, antimicrobial effect

## Abstract

Iron oxide nanoparticles (IONPs) have many practical applications, ranging from environmental protection to biomedicine. IONPs are being investigated due to their high potential for antimicrobial activity and lack of toxicity to humans. However, the biological activity of IONPs is not uniform and depends on the synthesis conditions, which affect the shape, size and surface modification. The aim of this work is to synthesise IONPs using a mixed method, i.e., chemical co-precipitation combined with biogenic surface modification, using extracts from spent hops (*Humulus lupulus* L.) obtained as waste product from supercritical carbon dioxide hop extraction. Different extracts (water, dimethyl sulfoxide (DMSO), 80% ethanol, acetone, water) were further evaluated for antioxidant activity based on the silver nanoparticle antioxidant capacity (SNPAC), total phenolic content (TPC) and total flavonoid content (TFC). The IONPs were characterised via UV-vis spectroscopy, scanning electron microscopy (SEM), energy-dispersive spectrometry (EDS) and Fourier-transform infrared (FT-IR) spectroscopy. Spent hop extracts showed a high number of flavonoid compounds. The efficiency of the solvents used for the extraction can be classified as follows: DMSO > 80% ethanol > acetone > water. FT-IR/ATR spectra revealed the involvement of flavonoids such as xanthohumol and/or isoxanthohumol, bitter acids (i.e., humulones, lupulones) and proteins in the surface modification of the IONPs. SEM images showed a granular, spherical structure of the IONPs with diameters ranging from 81.16 to 142.5 nm. Surface modification with extracts generally weakened the activity of the IONPs against the tested Gram-positive and Gram-negative bacteria and yeasts by half. Only the modification of IONPs with DMSO extract improved their antibacterial properties against Gram-positive bacteria (*Staphylococcus epidermidis*, *Staphylococcus aureus*, *Micrococcus luteus*, *Enterococcus faecalis*, *Bacillus cereus*) from a MIC value of 2.5–10 mg/mL to 0.313–1.25 mg/mL.

## 1. Introduction

There are many nanoparticles containing iron (Fe), including nano zero-valent iron (NZVI), oxides, hydroxides and oxyhydroxides of iron (II) and iron (III), e.g., Fe(OH)_3_, Fe(OH)_2_, ferrihydryt (Fe_5_HO_8_-4H_2_O), Fe_3_O_4_, FeO, five polymorphs of FeOOH and four of Fe_2_O_3_ [1]. The most common iron oxides that occur naturally are magnetite (Fe_3_O_4_), maghemite (γ-Fe_2_O_3_) and hematite (α-Fe_2_O_3_). Fe can also be present in nanomaterials in the form of nanoalloys or core–shell nanoparticles [2,3].

Depending on the type, iron nanoparticles (INPs) have found many different applications. Their electrical applications are mainly due to their high magnetism, good thermal and electrical conductivity and high microwave adsorption capacity [4,5]. Due to their catalytic activity and large surface area, they are used in catalytic reactions. As INPs are non-toxic and have high dimensional stability, they have found many biomedical applications, such as magnetic resonance imaging, drug delivery and gene therapy [4,6]. Magnetic and superparamagnetic iron oxide nanoparticles (IONPs) have been used for drug delivery in cancer treatments [7,8]. INPs have been used to remove organic and inorganic pollutants from water, soil and sediments in the natural environment [9,10,11]. Examples include the use of NZVI, which can destroy chlorinated organic hydrocarbons such as trichloroethylene [12], trichloroethene (TCE) [13,14,15] and dibenzo-p-dioxin [11]; reduce chlorinated ethanes [16]; and remove nitrites [17]. There is great interest in the possibility of using iron oxide nanoparticles to remove pollutants from the environment, e.g., phenol [18], oxyanions, including arsenite, arsenate, chromate, vanadate and phosphate, or remove toxic metal ions, e.g., for the adsorption of lead (II) [19] and arsenic [20,21,22,23,24,25,26]. An example is magnetic iron oxide modified with 1,4,7,10-tetraazacyclododecane (Fe_3_O_4_@SiO_2_-cyclen), which is able to selectively sorb heavy metal ions Cd^2+^, Pb^2+^ and Cu^2+^ [27].

The preparation of INPs, like other nanomaterials, can be achieved through “top-down methods”, in which the material is ground to nanosize [28,29,30,31,32], and “bottom-up methods” [33], in which the synthesis involves the self-organisation of atoms into new nuclei [31,34] as a result of aerosol, sol-gel processes, spinning, co-precipitation [35], mineralisation, sonochemical synthesis, microemulsion, etc. [4,34]. The use of a “bottom-up” method requires the purification of the nanoparticles and their separation from the reaction mixture [4,36].

NPs prepared using chemical and physical methods tend to form larger aggregates [37,38,39,40,41], which requires the addition of stabilisers in the form of polymers (PEG, polyacrylic acid, 4-butanediphosphonic acid and methoxyethoxyethoxyacetic acid (MEEA)) or surfactants [42]. Chitosan (CTS) is an example of a non-toxic, biocompatible, biodegradable natural polymer that can be used to coat magnetic nanoparticles. It is a hydrophilic polymer of a cationic nature that also has antibacterial properties [43,44,45].

Another possibility for bottom-up production is the so-called green synthesis, which is an attractive alternative to the above-mentioned methods due to its simplicity, cheapness and ecological safety using biological materials for the synthesis [34,46,47]. Green synthesis involves mixing appropriate precursors, most commonly aqueous solutions of iron salts (II) and (III), i.e., chlorides, sulphates (VI) and nitrates (V), in low concentrations ranging from 0.01 to 0.1 M [4,34,46,48], with material derived from bacteria, fungi, algae and plants [34,49,50]. The ecological synthesis of iron nanoparticles is preceded by the preparation of extracts containing bioactive compounds. It is known that many natural antioxidants and secondary metabolites, i.e., polyphenols, flavonoids, tannic acids, terpenoids, carboxylic acids, carotenoids, alkaloids, glycosides, vitamins and phenolic acids, have the ability not only to reduce iron ions but also to stabilise nanoparticles by binding to their surface [51,52,53,54].

The use of extracts has been described in the literature, such as the use of tea extract (polyphenols) as both a reducing and stabilising agent [55]. Iron oxide nanoparticles synthesised from clove and green coffee extracts showed a sorption capacity for divalent metal ions Cd^2+^ and Ni^2+^ and a broad spectrum of antibacterial activity against Gram-positive *Staphylococcus aureus* (*S. aureus*) and Gram-negative *Escherichia coli* (*E. coli*) bacteria [56]. Iron nanoparticles prepared from cloves had a MIC of 62.5 µg/mL against *S. aureus* and 125 against *E. coli* and were more effective against the tested pathogens than those prepared from green coffee with MIC values of 125 and 150 µg/mL, respectively [56]. Experimental studies have confirmed that superparamagnetic iron oxide nanoparticles obtained using this co-precipitation method, in which CTS protects the surface, have excellent antibacterial activity against the Gram-negative bacteria *Pseudomonas aeruginosa* (*P. aeruginosa*) and *E. coli* [57]. Spinach leaf extract and banana peel have also been considered for the preparation of iron nanoparticles [58]. The synthesised nanoparticles exhibited antimicrobial activity against two foodborne bacteria, i.e., *Bacillus subtilis* (*B. subtilis*) and *E. coli*, while remaining non-toxic against *Drosophila melanogaster* (*D. melanogaster*) in vivo.

In a study by Das et al. [59], *Humulus lupulus* extract was used for the first time for the green synthesis of silver nanoparticles, which showed antibacterial and anticancer activity with minimal genotoxicity and haemolysis.

Hops (*Humulus lupulus* L., Cannabaceae) are grown throughout the world as a raw material for beer production. The biological properties of this plant are related to the content of active compounds extracted from hop cones [60]. Hop cone extracts have antioxidant [61], antimicrobial and anti-inflammatory properties, as well as antimutagenic, antiallergic, neuroprotective and estrogenic properties [62,63]. The chemical compositions of the extracts vary and depend mainly on the variety and the growing environment [64]. The most important components of extracts for the industry are polyphenols, essential oils and alpha and beta acids. The phenolic fraction consists mainly of flavonoids, including xanthohumol [63] with high antimicrobial and antioxidant potential [65,66]. Also, α acids (humulones, i.e., cohumulone, humulone and adhumulone) and β-acids (lupulones, i.e., colupulone, lupulone and adlupulone) have confirmed antimicrobial activity [63]. α-acids are isomerised at high temperature to iso-α-acids, which also have antibacterial activity, mainly against Gram-positive bacteria [67,68]. Studies have confirmed the antibacterial, antiviral, antifungal and antiparasitic properties of compounds found in female cones and leaves [60,69,70]. Drug-resistant strains of *Staphylococcus aureus*, *Trypanosoma brucei* and *Leishmanis mexicana* were found to be sensitive to xanthohumol and lupulone administered in synergy with antibiotics [69,71].

Currently, the industrial production of hop extracts is carried out using the supercritical CO_2_ method (30 MPa, 50 °C) [72,73]. In Poland, this method is used by the Institute of Fertilisers in Puławy. The extracts are rich in soft and hard resins [74]. About 1000 tonnes of hop waste are produced annually, containing substances that are poorly soluble in non-polar supercritical CO_2_. Hops, therefore, contain valuable polar substances, i.e., proteins, amino acids, mineral salts, vitamins, proteins, sugars, polyphenols including prenylflavonoids (xanthohumol) and salts.

Spent hops left over from supercritical carbon dioxide extraction are treated as a source of xanthohumol, which has anti-cancer and antioxidant properties. Spent hops are also used as a feed additive. In a 2008 study [75], the total content of hop acids in extracted hop kernels was determined to be 1.051 µg/g (222 µg bitter isohumulones and 757 µg humulones per g). About 1000 prenylated flavonoids have been identified [76], occurring in several plant families that use these compounds in defence against pathogens and stress caused by unfavourable environmental conditions. Prenylated flavonoids include isoxanthohumol (IXN), xanthohumol (XN) and 8-prenylnaringenin (8PN), which are found in hops and hop waste remaining after extractions with supercritical carbon dioxide.

The increasing number of infections that are resistant to treatment with known antimicrobial agents stimulates the search for new compounds as alternatives to antibiotics [77,78,79,80]. The most dangerous are nosocomial infections caused by the formation of a bacterial biofilm composed of Gram-negative bacteria, namely *Pseudomonas aeruginosa* (*P. aeruginosa*) and *Escherichia coli* (*E. coli*) [81,82]. Taking into account the above needs and in cultivating a green approach to biotechnology, we designed a synthesis of iron oxide nanoparticles (IONPs) modified with the extract of wasted and unusable spent hops, which are a by-product of supercritical carbon dioxide extraction.

Not without significance is the fact that superparamagnetic iron oxide nanoparticles (SPIONPs), such as magnetite (Fe_3_O_4_), have antibacterial properties [83] and are biocompatible (BC) with the human body [1]. However, it is known that the observed activity of nanomagnetites is variable and depends on their size and shape and the type of stabilisers used [84]. Therefore, the synthesis procedure determines the final properties of the nanoparticles.

In this work, iron oxide nanoparticles (IONPs) modified with extracts from spent hops were obtained for the first time using a hybrid method, combining the chemical method of IONP synthesis through the co-precipitation of mixed ferrous and ferric salts from a solution in an alkaline medium with modification using plant extracts.

Initially, hop extracts were tested for their antioxidant activity. The final product was tested for antimicrobial activity against Gram-positive and Gram-negative bacteria and yeasts.

## 2. Results

### 2.1. Spent Hop Extract Characteristics

The total polyphenols and flavonoids, expressed as vitamin C equivalents, are summarised in Table 1. Regarding the content of each group of compounds, all types of extracts showed a greater amount of total flavonoid compounds than polyphenols. The potency of the extraction solvents used can be ranked from the most effective to the least effective for the extraction of flavonoids as follows: DMSO > 80% ethanol > acetone > water.

The total antioxidant capacity (TAC) should be understood as a net measure of all redox substances present in biological materials [85]. The TAC for complex mixtures is additive.

In order to compare TAC values for different samples using the chosen assay, it is necessary to consider the range of concentrations at which a linear relationship occurs.

The evaluation of the TAC using the SNPAC method is based on the plotting of a calibration curve for the antioxidant, i.e., the relationship between absorbance and concentration. The molar absorbance (ε) is estimated from the slope of the calibration curve.

The samples were prepared by mixing an initial solution of AgNPs with the citrate capping agent with different volumes of the standard or extracts (5–795 µL) and water. Figure 1 shows the changes in absorbance of the mixture with an increasing volume of the extract or standard. The absorbance value corresponding to the plasmon resonance of the AgNPs was linearly dependent on the antioxidant concentration/volume. It can be seen that the reducing potential of all mixtures increases, indicating a high concentration of antioxidants [86,87,88].

The plasmon absorbances of the AgNPs were perfectly linear (R^2^ > 0.99), but in different ranges depending on the type of extraction solvent, i.e., from 5 to 100 µL for all extracts except DMSO extract, from 5 to 20 µL of extract for DMSO extract and from 5 to 50 µL for the antioxidant standard (vit. C) with a concentration of 1 mg mL^−1^, i.e., in the concentration range from 10,139 × 10^−6^ to 101,386 × 10^−6^ M vit. C (final concentrations in the mixture and molar absorption capacity ε = 1.73 × 104 L mol^−1^ cm ^−1^).

The obtained relationships are in agreement with the observation of Eustis et al., who claimed that among the various factors influencing the absorption of SPR (surface plasmon resonance absorption) by AgNPs, the reaction stoichiometry, particle morphology and dielectric constants of the surrounding medium are crucial [89].

The addition of DMSO to the reaction system used in the SNPAC method affects the A423 nm versus volume of extract added. The inhibition of absorbance after the addition of 20 µL of DMSO extract is analogous to the observed slowing of the increase in the fluorescence of fluorescent probes in the vicinity of this solvent, as described by Setsukinai et al. [90]. The tighter solvation of silver cations by DMSO, combined with the strong Lewis basicity of this solvent and the low medium effect (−5.11) [91], slows down the reduction of silver ions. The silver cations in DMSO can form di- and tetrasolvated species [Ag(DMSO)2]+ and [Ag(DMSO)4]+, as reported by Ahrland et al. [92]. Rodríguez-Gattorno et al. confirmed that the reaction to form AgNPs in DMSO is slow and practically impossible without the addition of citrate as a reducing and stabilising agent [93]. The oxidation of DMSO leads to the formation of dimethylsulphone (CH_3_)_2_SO_2_. However, DMSO does not reduce various silver salts such as nitrate, perchlorate and metavanadate, even when heated to 80 °C. This reaction occurs only when trisodium citrate is added, which is responsible for the reduction of silver even at room temperature. Also, the ζ-potential, tested in different solvents, shows the highest value in water (−26.5 mV) and the lowest in DMSO (−15.8 mV) [94]. Furthermore, according to recent findings, the sizes of NPs such as magnetite particles decrease with increasing concentrations of DMSO [95].

The above findings may explain the unusual behaviour of the relationship between the absorbance and volume of extract obtained for DMSO compared to those obtained for extracts prepared with water and 80% ethanol. As can be seen in Figure 1, despite the rich content of flavonoids and polyphenols in the extract prepared with DMSO, the reduction of silver ions is inhibited after the addition of 20 µL of DMSO.

With regard to the total antioxidant capacity (TAC) of the extracts evaluated via SNPAC, the highest reducing power was observed for the DMSO extract, followed by water and 80% ethanol, and the lowest, for the acetone extract. The vitamin C equivalent antioxidant capacity (TAC) of a given extract is the ratio of its absorbance from the linearity region (observed after the addition of 15 µL of extract) to the molar extinction coefficient (ε) value of vitamin C under the same SNPAC test conditions [96].

### 2.2. Humulus Lupulus–Iron Oxide NP Characteristics

#### 2.2.1. UV-Vis Analysis

As a result of the chemical synthesis, nanoparticles were created, which were visible as a colour change from yellowish to intense black. Under the influence of an external magnetic field, the nanoparticles could be separated from the solution (Figure 2).

The UV-vis spectra recorded in the 330–600 nm range are shown in Figure 3. A spectrophotometric analysis of the ferric chloride solution showed characteristic absorption bands around 340 nm. The reduction of iron was confirmed using UV-vis spectra and is shown in Figure 3a. The reaction product is visible as the colour of the reaction mixture changes from yellow orange to dark black. In addition, the UV-vis spectra after the reaction showed a broad absorption at a higher wavelength and no sharp absorptions at lower wavelengths. Similar results have been obtained by other researchers [55,97,98,99,100,101,102,103]. A slight variation in the wavelengths of the peaks visible in the spectrum may be due to different measurement conditions, different materials used for synthesis, etc. Figure 3b shows the absorption spectra of the extracts before and after sorption with IONPs. As can be seen, the UV-vis spectrum of the extract has a broad band below 330 nm, confirming the high content of polyphenolic compounds [86]. After the addition of IONPs, a loss of colour of the extract and a decrease of absorbance in the range of 330–600 nm were observed. The decrease in the absorbance of the extract after the addition of IONPs confirms the sorption of the active components of the extract by the nanoparticles.

#### 2.2.2. FT-IR Measurements

The FT-IR/ATR spectra of the studied samples, i.e., aqueous and alcoholic spent hop extracts, pristine IONPs and IONPs with adsorbed aqueous and alcoholic extracts, are presented in Figure 4.

An analysis of the FT-IR spectrum of the IONPs confirmed that they are iron oxide nanoparticles [104]. The spectrum of pristine IONPs shows well-defined peaks at 3425, 1631, 1342, 938, 620 and 547 cm^−1^. The two peaks at 620 and 547 cm^−1^ result from the presence of iron-oxygen (Fe–O) bonds. The peaks at 3425 and 1631 cm^−1^ are the result of bending vibrations of hydroxyl –OH groups and adsorbed water, respectively. The bands at 1342, ~1090, 938 and 830 cm^−1^ may indicate the presence of nitrate groups (precursors of iron ions).

Hops contain more than 1000 various chemical substances, i.e., flavonoids, volatile oils, hop acids and proteins [105,106]; hence, the compositions of both the aqueous and alcoholic hop extracts are complex [107,108,109]. In analysing the FT-IR/ATR spectra of the aqueous and alcoholic extracts, it can be concluded that their chemical compositions are similar, but not identical. Bands with maxima at ~3305 and 3195 cm^−1^ represent the stretching vibrations of both the hydroxyl –OH groups and N–H in amino groups. Stretching vibrations of aliphatic C–H were observed within 2960–2853 cm^−1^ and ~1440–1350 cm^−1^. These bands indicate the presence of i.a. polysaccharides, proteins and phenolic compounds in the hop extracts.

The bands at 1738 and 1712 cm^−1^ indicate the presence of C=O (carboxylic acids, esters, aldehydes and ketones), while the band at 1671 cm^−1^ may indicate both C=O stretching vibrations in flavones and quinones, amides (C=O in amide I) and C=C stretching and C=N stretching vibrations [110]. The band at 1607 cm^−1^ may be responsible for C=N and C=C (also a ring stretching vibration in aromatic structures). However, the band at ~1607 cm^−1^ in the extract spectra may also be assigned to asymmetric COO– stretching, while the peak at ~1392 cm^−1^ may point to COO– symmetric stretching. The latter peak can also be assigned to C–OH stretching vibrations of the phenolic and/or alcoholic groups. Peaks within 1270–1230 cm^−1^ may be responsible for C–O stretching vibrations in the phenols. Bands within the 1100–1030 cm^−1^ range are attributed to C–O–C stretching vibrations of aromatic ethers and carbohydrates. The presence of carbohydrates is also indicated by the peak at 991 cm^−1^ of CH_2_OH in carbohydrates (shoulder, visible only in the alcoholic extract spectrum). The peak at ~1137 cm^−1^ may be attributed to the C–N stretching vibration of aromatic primary and secondary amines, and the bands within 930–600 cm^−1^ correspond to primary and secondary amines and amides (–NH_2_ wagging) and/or C–O and C–O–C symmetric stretching. The bands at ~591 cm^−1^ and ~518 cm^−1^ are C–CO–C and C–CO in-plane deformation vibrations in the ketone groups, respectively [110]. In summary, all these bands indicate the presence of flavonoids such as xantohumol and/or isoxantohumol [105] as well as bitter acids (i.e., humulones, lupulones) and proteins in the extracts.

As it was mentioned, the FT-IR/ATR spectra of aqueous and alcoholic extracts are similar, although there are some differences in the positions and intensities of the peaks. In the spectrum of the alcoholic extract, the peaks of the C=O bands at 1738 and 1712 cm^−1^ have much higher intensities, which may indicate a higher amount of carboxylic acids, esters, aldehydes and ketones. The peaks of aliphatic C–H groups within 2960–2853 cm^−1^ also have higher intensities.

The adsorptions of both the alcoholic and aqueous extracts in the IONPs are visible in the form of bands that were absent in the spectrum of pure IONPs (Figure 4). In the spectrum of IONPs with adsorbed *Humulus lupulus* aqueous extract, bands for the hydroxyl –OH groups and N–H in amino groups (3365–3195 cm^−1^) and C–H groups appear (2960–2853 cm^−1^). The C–H bands have higher intensities in the spectrum of IONPs with adsorbed aqueous extract. The adsorption of the extracts is also evidenced by bands with relatively high intensities at 1607, 1370 and 1075 cm^−1^. In the spectrum of IONPs with adsorbed aqueous extract, there are additional bands absent or bands that have lower intensities in the spectrum with adsorbed alcoholic extract, namely 1269, 1033 and bands in the 930–600 cm^−1^ range. The intensities of the bands in the spectrum of IONPs with adsorbed aqueous extract are higher than in the case of IONPs with adsorbed alcoholic extract, which may indicate a greater adsorption of compounds from the aqueous extract. Nevertheless, the analysis of the FT-IR/ATR spectra allows us to confirm the adsorption of compounds present in both the aqueous and alcoholic extracts on the IONPs.

#### 2.2.3. Scanning Electron Microscopy (SEM) with Energy-Dispersive X-ray Spectroscopy (EDX)

SEM images of the IONPs obtained via chemical synthesis using the co-precipitation method showed a granular, uniform, spherical structure with a diameter ranging from 81.16 nm to 142.5 nm, as shown in Figure 5a. An initial measurement of the sizes of the nanoparticles using the Malvern Zetasizer Nano ZS with the DLS technique showed a size of 293.13 nm ± 18.50 with a PDI of 0.58 ± 0.09 (Figure 6). This higher value is probably due to the aggregation of the NPs in an aqueous environment visible on the SEM image (Figure 7a).

However, it can be seen that the synthesised IONPs are more clearly separated and can be better distinguished. After modification with the extract, the NPs appear to be more viscous and aggregated (Figure 7b). This change is reflected in the size of the NPs measured via DLS (Figure 6). As can be seen, after the modification, the size increases almost twice that of an average value of 544.66 21.02 nm. In addition, EDS was used to provide information on the chemical compositions of the identified nanostructures. The amplitude of the spectrum showed the presence of several characteristic iron peaks in the range of about 0.5 to 7 keV, as shown in Figure 5b.

### 2.3. Antimicrobial Activity

The antimicrobial activities of the extracts were tested against selected Gram-positive and Gram-negative bacteria and yeasts. The bioactivity results are presented in Table 2 as minimum inhibitory concentration (MIC) and minimum bactericidal concentration (MBC) values. Water extract showed weak bioactivity (MIC 5–10 mg/mL) against the tested reference strains, whereas ethanol extract showed moderate bioactivity against Gram-positive bacteria. The DMSO extract showed the highest activity against Gram-positive bacteria.

IONP surface modification did not increase the bioactivity of the new preparations. The antifungal (against *C. albicans*, *C. parapsilosis* and *C. glabrata*) and antibacterial bioactivities of the modified IONPs were weak and usually lower or equal to those of the extracts themselves.

Only surface modification with DMSO extract improved the activity of the IONPs against Gram-positive bacteria. While unmodified IONPs have MIC values in the range of 2.5–10 mg/mL, IONPs modified with DMSO extract have MIC values many times lower in the range of 0.313–1.25 mg/mL.

## 3. Discussion

Sergey V. Gudkov tried to answer the question on whether IONPs have significant antibacterial properties in his review article in 2021 [76]. The collected research confirms the antimicrobial effect of IONPs on Gram-negative and Gram-positive bacteria and fungi, with low toxicity toward eukaryotic cells. However, the effect of IONPs varies greatly depending on the type of microorganism and the method of synthesis. A practical inhibition of bacterial growth occurs at concentrations ranging from 1–3 µg/mL against *S. mutans* [111] through 3 mg/mL against *S. aureus* [80] to 62.5 mg/mL against *Corynebacterium* sp., *K. pneumonia*, *P. aeruginosa* and *S. aureus* [112]. However, it should be emphasised that in these cases, the IONPs were obtained via biogenic synthesis or modified with diethyl lenglycol or polyvinyl alcohol (PVA).

In our work, IONPs were obtained through chemical co-precipitation. Extracts from the waste products of supercritical carbon dioxide hop extraction were used to modify the IONPs for the first time. It seems that the observed low bioactivity of IONPs is related to the phenomenon of the agglomeration of magnetic nanoparticles, which does not disappear after modification with hop extracts, except for the DMSO extract. It has been repeatedly reported that magnetic nanoparticles (mainly magnetite and maghemite) tend to form aggregates or agglomerates due to a combination of van der Waals and magnetic forces [113]. Such associations can be permanent or reversible and largely determine the properties of magnetic nanoparticles. It should be emphasised that these processes are still under investigation and not fully understood [114]. As emphasised by many authors, the sizes of aggregated clusters can even exceed 1 μm [115,116]. Therefore, the formation of clusters affects both the transport of nanoparticles and the reactivity by changing the activity of a single nanoparticle.

Not only aggregation, but also the type of materials chosen to coat IONPs fundamentally determine their cytotoxicity [117]. In our work, the surface modification of IONPs using water and 80% ethanol extracts reduced or did not change the activity of the IONPs against Gram-positive and Gram-negative bacteria and yeasts. IONPs have activity comparable to that of aqueous extract with a MIC of 5 mg/mL toward Gram-positive bacteria, whereas its activity against Gram-negative and yeasts was comparable to that of ethanol extract. IONPs modified with ethanol extract showed anti-yeast activity comparable to that of aqueous extract with a MIC value of 2.5 to 5 mg/mL. However, it should be noted that the modification reduces the activity by half of the anti-yeast activity of unmodified IONPs.

*C. parapsilosis* and *C. albicans* live on the surface of the skin and are the most common cause of infections in hospital intensive care units. Similar to other Candida microorganisms, their main virulence factor is the ability to colonise artificial surfaces. The ability to colonise is determined by the ability to grow in the form of a biofilm. Therefore, unmodified IONPs have the potential to be used in anti-yeast preparations for topical applications. Furthermore, taking into account the fact that IONPs possess magnetic abilities, this can be used to hold them on artificial surfaces to protect against colonisation.

In our work, the ability of IONPs to inhibit the growth of Gram-positive bacteria (*S. epidermidis*, *S. aureus*, *M. luteus*, *E. faecalis*, *B. cereus*) increases from a MIC value of 2.5–10 mg/mL to 0.313–1.25 mg/mL after their modification with DMSO hop extract. This seems to be reasonable since the DMSO spent hop extract showed the highest values of TPC, TFC and TAC evaluated using the SNPAC method.

## 4. Materials and Methods

### 4.1. Materials

Ascorbic acid (vitamin C, vit. C), silver nitrate (AgNO_3_) and dimethyl sulfoxide (DMSO) were purchased from Sigma-Aldrich Inc., St. Louis, MO, USA. Ethanol was purchased from E.Merck (Darmstadt, Germany). Water with a resistivity of 18.2 MΩ cm was obtained from an ULTRAPURE Millipore Direct-Q 3UV-R (Merck, Darmstadt, Germany).

### 4.2. Extraction of Spent Hops

Hop cones (*Humulus lupulus*) were grown in the Lublin region (Chmielnik near Lublin). Spent hops (of the Marynka variety), which were subjected to supercritical CO_2_ extraction, were obtained from the Fertiliser Research Centre of the Institute of New Chemical Synthesis of the Łukasiewicz Research Network in Puławy, Poland. The spent hops were extracted with different solvents, i.e., water, 80% ethanol and DMSO. For the extraction, 5 g of dried plant material was suspended in 100 mL of solvent in a 250 mL Erlenmeyer flask and sonicated for 60 min in an ultrasonic bath (ultrasonic power 1200 W, frequency 35 kHz), a Bandelin Sonorex RK 103 H (Bandelin Electronics, Berlin, Germany), at 80 °C. After cooling, the extracts were centrifuged at 11,000 rpm for 15 min to precipitate traces of solids from the extract. The supernatants were collected, filtered through Whatman No. 1 filter paper and evaporated under vacuum. The residue was dissolved in 5 mL of water.

### 4.3. Total Flavonoid Content (TFC)

The total flavonoid content (TFC) was determined spectrophotometrically using a Genesys 20 spectrophotometer (The ThermoSpectronic, Waltham, MA, USA) according to Lamaison and Carnat [118]. Briefly, 1.0 mL of 1.2% aluminium chloride (AlCl_3_) in water was mixed with 100 µL of extract and 900 µL of water. Absorbance at 415 nm was measured against a blank (water) after 30 min.

The total flavonoid content was expressed as the mg reference equivalent (vit. C) per 1 mL^−1^ of extract. Quantification was evaluated using a linear curve of ascorbic acid used as a standard at concentrations ranging from 0.0625 to 1.00 mg mL^−1^. The calibration curve was generated from six standard solutions with concentrations of 0.0625, 0.125, 0.25, 0.5, 0.75 and 1.00 mg mL^−1^. The equation of the calibration curve was as follows:1.837 (±0.065)x − 0.029 (±0.036), R^2^ = 0.9950, F = 800.64, se = 0.05(1)

### 4.4. Total Phenolic (TPC)

The total phenolic content (TPC) was determined using the Folin–Ciocalteu method [119]. Briefly, 250 µL of extracts diluted 1:10 and 1000 µL of Folin–Ciocalteu phenol reagent diluted 1:10 (Sigma-Aldrich, St. Louis, MO, USA) were added to flasks and mixed thoroughly. Then, 1.0 mL of 10% Na_2_CO_3_ solution was added, and the mixture was kept in the dark for 2 h at room temperature. The absorbance was then read at 765 nm.

The total phenolic content was expressed as milligrams of the reference equivalent (vit. C) per 1 mL^−1^ of extract. Quantification was evaluated using a linear curve of the ascorbic acid used as the standard at concentrations ranging from 3.9063 to 125.00 µg mL^−1^. The calibration curve was constructed from six standard solutions with concentrations of 3.91, 7.81, 15.63, 31.25, 62.50 and 125.00 µg mL^−1^. The equation of the calibration curve was as follows:7.6201 (±0.2136)x + 0.0707 (±0.126), R^2^ = 0.9969, F = 1272.60, se = 0.02(2)

### 4.5. Silver Nanoparticle Antioxidant Capacity (SNPAC)

SNPAC measurements were performed according to Özyürek et al. [96]. The procedure has been described in detail previously [86,87]. The initial solution of silver nanoparticles (AgNPs) was prepared by mixing 5 mL of 1% aqueous tripotassium citrate solution with 50 mL of 1 mM AgNO_3_ at an elevated temperature (around 90 °C). The test samples were prepared by mixing 2 mL AgNPs, standard or tested extract (x) and (0.8 − x) mL water. After storage in the dark for 30 min, the mixtures were measured spectrophotometrically at 423 nm, which is characteristic of the surface plasmon resonance of AgNPs [87]. The calibration curve showed the relationship between the absorbance (A) and the final micromolar concentration of the standard antioxidant (vit. C). The calibration curve was linear from 10.16 to 101.39 µM (10.139; 20.277; 30.416; 40.554; 101.386 µM). The equation for the calibration curve was as follows:A_423nm_ = 0.0173 (±0.00075)c_µM_ + 0.112 (±0.039), R^2^ = 0.9944, F = 537.77, se = 0.05(3)

The total antioxidant capacity in µM (TAC) of the extracts was calculated by dividing the observed absorbance at λ = 423 nm for 15 µL of added sample by the molar absorptivity (ε) of vitamin C, which can be obtained from the slope of the calibration curve. The method is effective for antioxidants with a standard potential of less than 0.8 V due to E Ag(I), Ag° = 0.8 V.

### 4.6. Synthesis of IONPs

0.1 M FeCl_3_ and 0.1 M FeSO_4_ were mixed in a volume ratio of 2:1. 25 mL of 25%. NH_3_ was added dropwise to the solution with constant stirring. A black precipitate formed. It was separated using an external magnetic field and washed several times with deionised water until the solution above the precipitate reached pH 7. The IONPs were prepared according to the following reaction:FeCl_2_ × 4H_2_O + 2FeCl_3_ × 6H_2_O → 8NaCl + Fe_3_O_4_ + 14H_2_O(4)

### 4.7. Modification of the NP Surfaces Using Plant Extracts

The dried extracts (aqueous and 80% EtOH) were dissolved in 5 mL of water. Then 3 g of dried IONPs were added and conditioned at room temperature for 3 days; after which, the IONPs were separated using a magnet, and the supernatant was removed. The modified nanoparticles were washed several times with small volumes of deionised water and allowed to dry. The powders were then stored in airtight glass vials at 4 °C until further analysis.

### 4.8. Characterization of AgNPs

#### 4.8.1. UV-Vis Spectroscopy, SEM, EDS and DLS

IONPs were characterised spectrophotometrically using a Genesys 20 spectrophotometer (The ThermoSpectronic, Waltham, MA, USA).

The morphology and chemical composition of the IONPs were investigated using a Quanta 250 FEG scanning electron microscope from FEI (Almelo, The Netherlands) equipped with a FEG electron source and energy-dispersive X-ray spectroscopy (EDS). The experiment was carried out in the same conditions as previously described [87]. Observations were made using accelerating voltages ranging from 10 to 15 keV.

The Malvern Zetasizer Nano ZS (Malvern Instruments Ltd. (Malvern, UK), GB) with the DLS technique was used to measure the size of the nanoparticles.

#### 4.8.2. FT-IR

Fourier-transform infrared–attenuated total reflectance (FT-IR/ATR) spectra were recorded in the 3700–400 cm^−1^ range, resolution 4 cm^−1^, at room temperature using a Nicolet 6700 spectrometer (Thermo Fisher Scientific Inc., Waltham, MA, USA) and a Meridian Diamond ATR accessory (Harrick Scientific Products, Inc., Pleasantville, NY, USA). Interferograms of 512 scans were averaged for each spectrum. Dry potassium bromide (48 h, 105 °C) was used as a reference material to collect the ATR spectra. No smoothing functions were applied. All spectral measurements were performed at least in triplicate. Raw spectra were processed using OMNIC^TM^ software (Thermo Fisher Scientific Inc., USA) version 8.2.387.

### 4.9. Antimicrobial Activity Assay

Suspensions of all compounds in DMSO were screened for antibacterial and antifungal activities using the 2-fold microdilution broth method. The detailed procedure of the assay was previously described [120]. The minimal inhibitory concentrations (MIC) of the tested compounds were evaluated for the following panel of reference yeasts: *Candida parapsilosis* ATCC 22019, *Candida glabrata* ATCC 90030 and *Candida albicans* ATCC 102231. The panel of reference Gram-positive bacteria included *Staphylococcus epidermidis* ATCC 12228, *Staphylococcus aureus* ATCC 25923, *S. aureus* ATCC BAA-1707, *Micrococcus luteus* ATCC 10240, *Enterococcus faecalis* ATCC 29212 and *Bacillus cereus* ATCC 10876. The Gram-negative bacteria panel included *Escherichia coli* ATCC 25922, *Proteus mirabilis* ATCC 12453, *Salmonella Typhimurium* ATCC 14028, *Pseudomonas aeruginosa* ATCC 9027 and *Klebsiella pneumoniae* ATCC 13883.

## 5. Conclusions

This work describes for the first time the possibility of obtaining IONPs using a method that combines chemical synthesis with modification using plant extracts. The waste product remaining after the supercritical carbon dioxide extraction of hop cones was used to obtain the extracts. Among the different solvents used for extraction, the modification of IONPs with DMSO extract improved their antibacterial properties against Gram-positive bacteria almost tenfold, from a MIC of 2.5–10 mg/mL to 0.313–1.25 mg/mL. This is the first study to use spent hop extract to modify IONPs. The results obtained show the great potential of the waste material (spent hops), which should be further investigated for use in the biogenic synthesis of nanoparticles of other metals and their activity toward microorganisms as well as cancer cell lines.

## Figures and Tables

**Figure 1 antibiotics-13-00111-f001:**
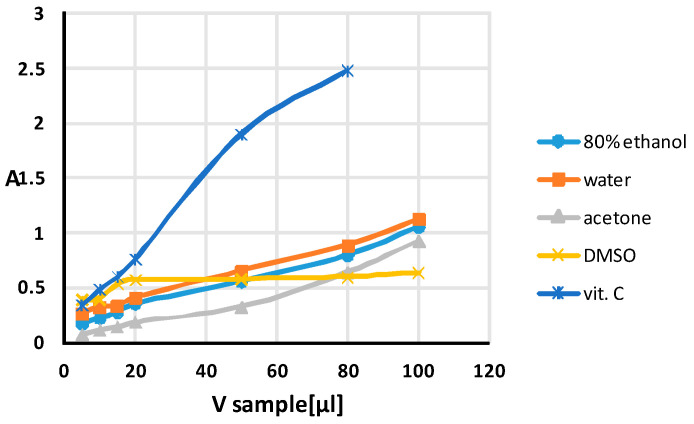
Absorbance (A) measured at 423 nm versus the volume of the extracts or standard antioxidant (vit. C) added to the test tubes with the initial solution of AgNPs with the citrate capping agent.

**Figure 2 antibiotics-13-00111-f002:**
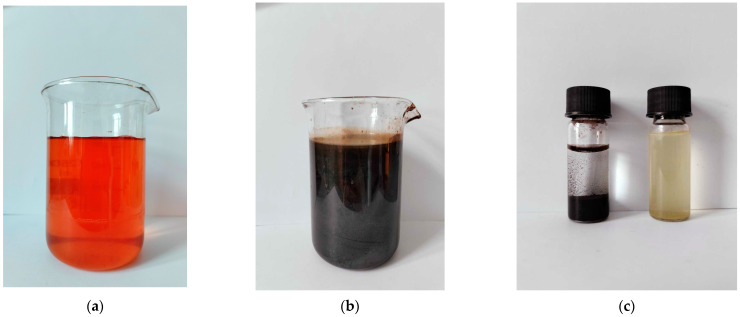
The visible colour change of the precursor solution FeCl_3_ (**a**) into synthesised IONPs (**b**). The prepared IONPs were added to an aqueous solution of plant extract and separated using a magnet (**c**).

**Figure 3 antibiotics-13-00111-f003:**
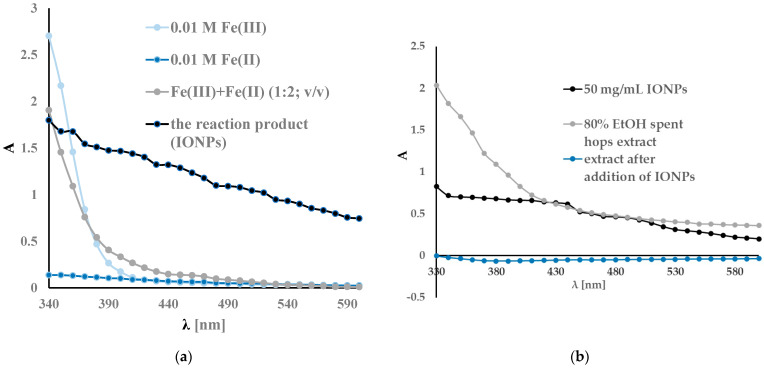
The UV-vis spectra recorded in the range of 330–600 nm during chemical synthesis: initial iron ion solution and IONPs (**a**) and during the modification of the IONPs using plant extract (**b**).

**Figure 4 antibiotics-13-00111-f004:**
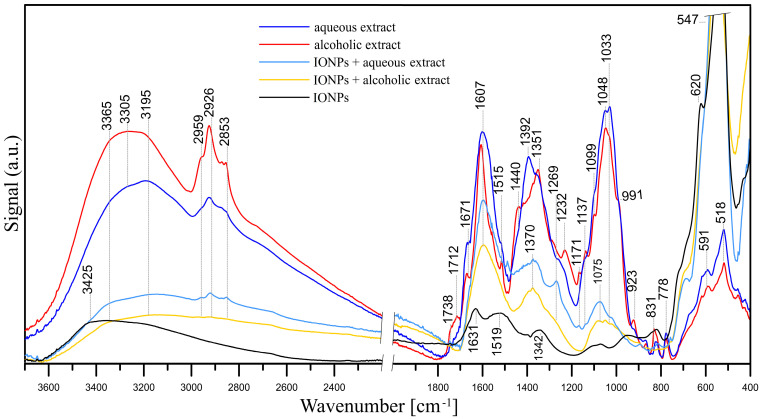
FT-IR/ATR spectra of studied samples: spent hop aqueous and alcoholic extracts, IONPs and IONPs with adsorbed *Humulus lupulus* aqueous and alcoholic extracts.

**Figure 5 antibiotics-13-00111-f005:**
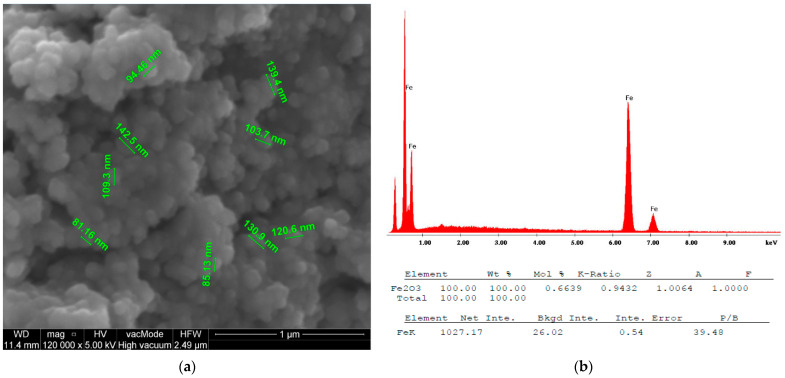
SEM image (**a**) with EDS analysis (**b**) of IONPs.

**Figure 6 antibiotics-13-00111-f006:**
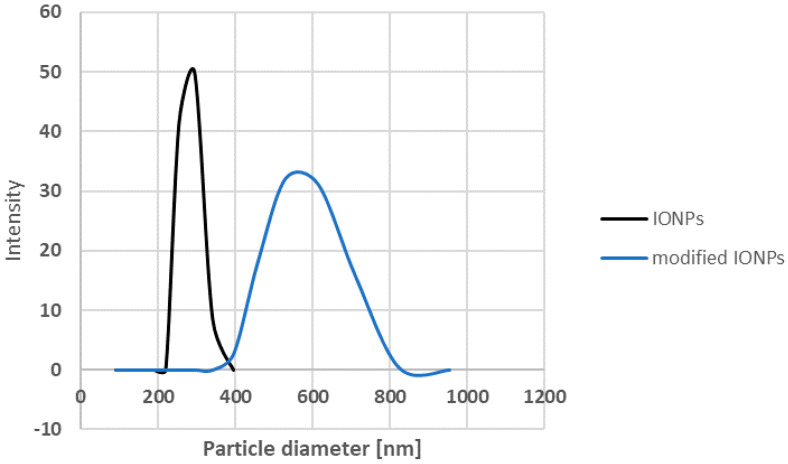
The hydrodynamic particle diameter of biosynthesised IONPs.

**Figure 7 antibiotics-13-00111-f007:**
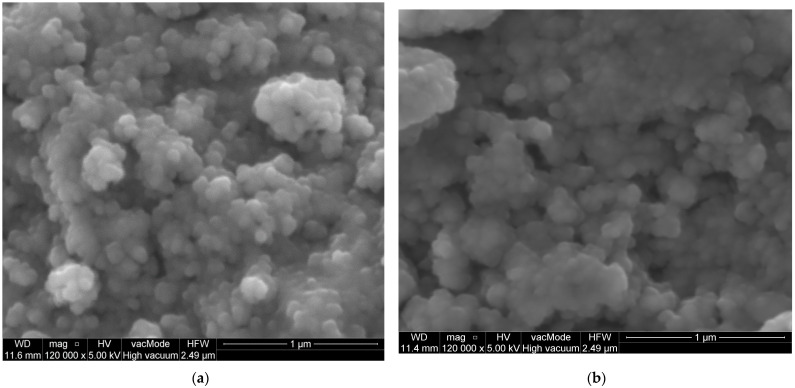
SEM images of IONPs (**a**) and IONPs modified using *Humulus lupulus* aqueous extract (**b**).

**Table 1 antibiotics-13-00111-t001:** Total polyphenols and flavonoids and the total antioxidant capacity (TAC) values expressed as mg mL^−1^ or mM of the vit. C equivalent obtained using the SNPAC method for the *Humulus lupulus* L. extracts.

Parameter	Extraction Solvent			
	80% EtOH	Acetone	DMSO	H_2_O
	TPC			
Abs.Mean (*n* = 3)	0.8823	0.6763	0.8983	0.3843
Std. Dev.	0.0031	0.0021	0.0006	0.0012
mg vit. C mL^−1^	0.0296	0.0247	0.0655	0.0140
	TFC			
Abs.Mean (*n* = 3)	0.8757	0.8233	1.2173	0.5077
Std. Dev.	0.0006	0.0025	0.0029	0.0006
mg vit. C mL^−1^	0.0247	0.0232	0.0340	0.0146
	SNPAC			
Abs.Mean (*n* = 3)	0.2772	0.1404	0.5301	0.3260
Std. Dev.	0.0021	0.0061	0.0012	0.0002
TAC (µM)	16.012	8.092	30.636	18.844

Abbreviations: vit. C—ascorbic acid equivalents; TAC—the total antioxidant capacity expressed as vit. C equivalents; EtOH—ethanol; Std. Dev.—the standard deviation of three independent measurements; Abs.—absorbance measured spectrophotometrically at 423 nm.

**Table 2 antibiotics-13-00111-t002:** Antimicrobial activity of the spent hop extracts, IONPs and spent hop-modified IONPs obtained via mix-mode chemical/biogenic synthesis presented as minimal inhibitory concentration (MIC) and minimal bactericidal concentration (MBC) values in mg L^−1^ or as the dissolution ratio of the initial extract in the case of DMSO.

Microorganism	IONPs + Water Extract		IONPs + 80% Ethanol Extract		IONPs + DMSO Extract		IONPs		Water Extract		80% Ethanol Extract		DMSO Extract	
	MIC	MBC	MIC	MBC	MIC	MBC	MIC	MBC	MIC	MBC	MIC	MBC	MIC	MBC
Gram-positive bacteria														
*Staphylococcus aureus* ATCC 25923	>10	Nd	>10	Nd	1.25	5	5	>10	5	10	0.313	0.625	1:3200	1:400
*Staphylococcus aureus* ATCC BA1707	>10	Nd	>10	Nd	0.625	2.5	>10	>10	5	10	0.313	0.625	1:6400	1:800
*Staphylococcus epidermidis* ATCC 12228	10	>10	10	>10	1.25	5	5	>10	5	10	0.156	2.5	1:3200	1:100
*Micrococcus luteus* ATCC 10240	10	>10	2.5	10	0.625	0.625	2.5	5	0.313	2.5	0.078	0.078	1:6400	1:200
*Bacillus cereus* ATCC 10876	>10	Nd	10	>10	0.313	0.313	5	>10	5	>10	0.156	5	1:12,800	1:3200
*Enterococcus faecalis* ATCC 29212	>10	Nd	5	>10	0.625	10	5	>10	10	>10	5	10	1:12,800	1:100
Gram-negative bacteria														
*Salmonella typhimurium* ATCC 14028	10	>10	10	>10	>10	Nd	5	>10	>10	>10	5	>10	>1:5	Nd
*Escherichia coli* ATCC 25922	>10	Nd	>10	Nd	>10	Nd	5	>10	5	10	5	5	>1:5	Nd
*Proteus mirabilis* ATCC 12453	>10	Nd	>10	Nd	>10	Nd	>10	>10	5	10	5	10	>1:5	Nd
*Klebsiella pneumoniae* ATCC 13883	10	>10	>10	Nd	>10	Nd	5	>10	5	5	5	2.5	>1:5	Nd
*Pseudomonas aeruginosa* ATCC 9027	10	>10	10	>10	>10	Nd	5	10	5	5	5	5	>1:5	Nd
Yeasts														
*Candida glabrata* ATCC 90030	10	10	10	10	10	10	5	5	5	5	5	5	1:10	1:10
*Candida albicans* ATCC 102231	5	10	5	10	10	10	2.5	5	5	5	2.5	5	1:20	1:10
*Candida parapsilosis* ATCC 22019	10	10	5	10	10	10	2.5	5	5	5	2.5	5	1:80	1:10

## Data Availability

The data are available for request from Prof. J. Flieger.

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
