# Peer review of "Characteristics and Antimicrobial Activities of Iron Oxide Nanoparticles Obtained via Mixed-Mode Chemical/Biogenic Synthesis Using Spent Hop (Humulus lupulus L.) Extracts"

_antibiotics, 2024, doi:10.3390/antibiotics13020111_

Round 1

Reviewer 1 Report

Comments and Suggestions for Authors

Antibiotics-2814710

Characteristics and Antimicrobial activities of iron oxide nanoparticles obtained in mixed mode chemical/biogenic synthesis method using spent hops (Humulus lupulus L.) extracts

Title: Very expressive

Abstract:

Page 1 lines 26-28: “The IONPs were characterized using UV-vis spectroscopy, scanning electron microscopy (SEM), energy dispersive spectrometry (EDS), Fourier transform infrared (FT-IR) spectroscopy”. NOT “The IONPs were characterized according to their antimicrobial and antifungal bioactivity and structure using UV-vis spectroscopy, scanning electron microscopy (SEM), energy dispersive spectrometry (EDS), Fourier transform infrared (FT-IR) spectroscopy.”

The abstract need to observe the results of antimicrobial activity and abstract focus on the NPs characterization and chemical profile.

Introduction: Prepared well

Results and Discussion: Prepared well

Where is the novelty of this study?

(Extraction of Spent Hops, chemical profile determination, synthesis of IONPs, characterization of IONPs, and antimicrobial activity assay)…. Then so what?

Materials and Methods

Page 13 line 451: 4.6. Characterization of AgNPs…..what is this? Just copy and paste from another work.

4.7. Antimicrobial activity assay

Suspensions of all compounds in DMSO were screened for antibacterial and antifungal activities by 2-fold microdilution broth method. The detailed procedure of assay was previously described [111]. Where is the reference [111] in the reference list?, the references ended with reference No. 110.

There are major written errors in materials and method part need to rewrite

Also, Conclusion: need to rewrite

References: in to check

Comments on the Quality of English Language

Extensive editing of English language required

Author Response

Characteristics and Antimicrobial activities of iron oxide nanoparticles obtained in mixed mode chemical/biogenic synthesis method using spent hops (Humulus lupulus L.) extracts

Title: Very expressive

Answer: Thank you very much for taking the time to review this manuscript. Please find the detailed responses below and the corresponding revisions/corrections highlighted/in the re-submitted files. Thank You.

Abstract: Page 1 lines 26-28: “The IONPs were characterized using UV-vis spectroscopy, scanning electron microscopy (SEM), energy dispersive spectrometry (EDS), Fourier transform infrared (FT-IR) spectroscopy”. NOT “The IONPs were characterized according to their antimicrobial and antifungal bioactivity and structure using UV-vis spectroscopy, scanning electron microscopy (SEM), energy dispersive spectrometry (EDS), Fourier transform infrared (FT-IR) spectroscopy.”

Answer: Thank you very much for this suggestion. The sentence has been corrected (the change is marked in red in the new version of the manuscript).

The abstract need to observe the results of antimicrobial activity and abstract focus on the NPs characterization and chemical profile.

Answer: Thank you for point it out. We added this information to abstract.

Introduction: Prepared well

Answer: Thank you very much for this opinion and for noticing our efforts.

Results and Discussion: Prepared well

Answer: Thank you very much for the reviewer’s positive opinion.

Where is the novelty of this study? (Extraction of Spent Hops, chemical profile determination, synthesis of IONPs, characterization of IONPs, and antimicrobial activity assay)…. Then so what?

                Answer: The aim of the study and its novelty were added at the end of Introduction part:

The growing number of infections resistant to treatment with known antimicrobial substances stimulates the need to search for new preparations as an alternative to antibiotics [95-98]. The most dangerous are nosocomial infections caused by the formation of a bacterial biofilm composed of Gram-negative bacteria. bacteria Pseudomonas aeruginosa (P. aeruginosa) and Escherichia coli (E. coli) [99,100].

Taking into account above needs and cultivate a green approach to biotechnology we design synthesis of iron oxide nanoparticles (IONPs) modified with extract of wasted, and unusable spent hops being a byproduct from supercritical carbon dioxide extraction

Not without significance was the fact that superparamagnetic iron oxide nanoparticles (SPIONPs), such as magnetite (Fe3O4), have antibacterial properties [58] and are bio-compatible (BC) with the human body [59]. However, it is known that the observed activity of nanomagnetites are different and depend on their size and shape and the type of stabilizers [60]. Therefore, the synthesis procedure determines the final properties of the nanoparticles.

In this work, iron oxide nanoparticles (IONPs) modified with extracts from spent hops were obtained by a hybrid method, for the first time, combining the chemical method of IONPs synthesis by co-precipitation of mixed iron (II) and (III) salts from a solution in an alkaline medium, with modification using plant extracts.

Materials and Methods: Page 13 line 451: 4.6. Characterization of AgNPs…..what is this? Just copy and paste from another work.

                Answer: Indeed, we used a similar method previously to analyze silver nanoparticles, so we shortened the description of methodology to the necessary information and cited the relevant literature.

4.7. Antimicrobial activity assay: Suspensions of all compounds in DMSO were screened for antibacterial and antifungal activities by 2-fold microdilution broth method. The detailed procedure of assay was previously described [111]. Where is the reference [111] in the reference list?, the references ended with reference No. 110.

                Answer: Thank you for pointing this out. We agree, this is our numbering error. Therefore, We have changed the Ref. number into Ref. 110.

There are major written errors in materials and method part need to rewrite

Answer: The manuscript was checked and corrected by a professional English translator.

Also, Conclusion: need to rewrite

Answer: The manuscript was checked and corrected as above.

References: in to check

Answer: The references were checked again regarding duplicates and entries consistent with the requirements of the journal. Numbering errors have been corrected.

Reviewer 2 Report

Comments and Suggestions for Authors

Characteristics and Antimicrobial activities of iron oxide nanoparticles obtained in mixed mode chemical/biogenic synthesis method using spent hops (Humulus lupulus L.) extracts.

The abstract is well written

INTRODUCION

The paragraph from line 67 to 90 is very long and tiresome to read, I suggest you split it into two paragraphs;

On line 95-105, write the genus name of the species of bacteria, since they were not mentioned before;

What is the significant importance of using Humulus lupulus L. in this study with part of the synthesis? It's not clear from the introduction

RESULTS

The paragraph from line 131-170 is very long and tiresome to read, I suggest you split it into two paragraphs;

Figure 4: The resolution is not good, it needs to be improved;

Table 2. Write out the genera of the species studied in full, do not delete.

DISCUSSION

In line 318-338, it looks like an introductory topic. I suggest improving the discussion of this manuscript.

Author Response

Characteristics and Antimicrobial activities of iron oxide nanoparticles obtained in mixed mode chemical/biogenic synthesis method using spent hops (Humulus lupulus L.) extracts.

The abstract is well written

Answer: The authors would like to thank the reviewer for his time and valuable comments, which allowed us to improve our work.

 INTRODUCION

The paragraph from line 67 to 90 is very long and tiresome to read, I suggest you split it into two paragraphs;

Answer: We agree with the reviewer that the paragraph is too long. The paragraph has been split into three.

On line 95-105, write the genus name of the species of bacteria, since they were not mentioned before;

Answer: Thank you, we agree that abbreviations should be explained the first time they are used. The fault has been corrected.

What is the significant importance of using Humulus lupulus L. in this study with part of the synthesis? It's not clear from the introduction

Answer: We added at the end of Introduction explanation:

The growing number of infections resistant to treatment with known antimicrobial substances stimulates the need to search for new preparations as an alternative to antibi-otics [95-98]. The most dangerous are nosocomial infections caused by the formation of a bacterial biofilm composed of Gram-negative bacteria. bacteria Pseudomonas aeruginosa (P. aeruginosa) and Escherichia coli (E. coli) [99,100]. Taking into account above needs and cultivate a green approach to biotechnology we design synthesis of iron oxide nanoparti-cles (IONPs) modified with extract of wasted, and unusable spent hops being a byprod-uct from supercritical carbon dioxide extraction

Not without significance was the fact that superparamagnetic iron oxide nanoparti-cles (SPIONPs), such as magnetite (Fe3O4), have antibacterial properties [58] and are bio-compatible (BC) with the human body [59]. However, it is known that the observed activi-ty of nanomagnetites are different and depend on their size and shape and the type of sta-bilizers [60]. Therefore, the synthesis procedure determines the final properties of the na-noparticles.

In this work, iron oxide nanoparticles (IONPs) modified with extracts from spent hops were obtained by a hybrid method, for the first time, combining the chemical method of IONPs synthesis by co-precipitation of mixed iron (II) and (III) salts from a solution in an alkaline medium, with modification using plant extracts.

RESULTS

The paragraph from line 131-170 is very long and tiresome to read, I suggest you split it into two paragraphs;

Answer: The long paragraph has been divided into several smaller paragraphs. Thank you for this suggestion.

Figure 4: The resolution is not good, it needs to be improved;

Answer: Thank You for pointed it out. Fig.4 has been improved.

Table 2. Write out the genera of the species studied in full, do not delete.

Answer: It was done.

DISCUSSION

In line 318-338, it looks like an introductory topic. I suggest improving the discussion of this manuscript.

Answer: Thank you for this suggestion. We transfer these part to introduction and improved discussion.

Reviewer 3 Report

Comments and Suggestions for Authors

In this manuscript, the Authors studied the chemical/biogenic synthesis of IONP and tested their antibacterial impacts on hop extracts.  UV-vis spectroscopy, scanning electron microscopy (SEM), 27 energy dispersive spectrometry (EDS), and Fourier transform infrared (FT-IR) spectroscopic measurements were studied for characterization. Unfortunately, I cannot recommend this manuscript for the "Antibiotics" journal as serious concerns exist about the synthesis method and the IONP product's reactivity against bacteria. I also have the following concerns:

1. UV-Vis spectroscopic data is useless in the presented version.

2. There is no proof for the formation of IO NP.

3. SEM images cannot prove the size. DLS may help.

4. HOP extracts have better antibacterial activity than IONP+HOP extracts (especially ethanol extracts), So, it is not useful for further processing.

Author Response

In this manuscript, the Authors studied the chemical/biogenic synthesis of IONP and tested their antibacterial impacts on hop extracts.  UV-vis spectroscopy, scanning electron microscopy (SEM), 27 energy dispersive spectrometry (EDS), and Fourier transform infrared (FT-IR) spectroscopic measurements were studied for characterization. Unfortunately, I cannot recommend this manuscript for the "Antibiotics" journal as serious concerns exist about the synthesis method and the IONP product's reactivity against bacteria. I also have the following concerns:

Answer: We accept this harsh assessment of our work. However, if there is an opportunity for discussion and any chance of convincing the reviewer of our vision, we would like to highlight a few strengths of our work:

  • Methods for the synthesis and analysis of IONPs are described and our work was guided by the experience of other authors. As is well known, there are many iron oxide nanoparticles. The conditions we described provided the following reaction to form black Fe3O4 [Ramimoghadam et al., 2014; Doan, 2023; Doan et al., 2023]. Fe2+ + 2Fe3+ + 8OH- → Fe3O4 + 4H2O
  • The study of the nature of the surface is the subject of our next paper, which is in preparation. There, we used other XPS methods and molecular modelling in addition to FTIR and SEM-EDS, DLS. This work is the subject of a patent, so we will publish it when the patent gets a registration number. However, to improve the current manuscript we have developed the synthesis section.
  • We have made some interesting observations about the use of DMSO as a solvent for the synthesis of nanoparticles. This is partly in line with the observations of other authors, but we believe that our observations and their interpretations, included in the paper, may be valuable to other researchers.
  • We are popularizing and testing, once again, on various examples, the SNPAC nanoparticle method for the determination of antioxidant activity. Our experience shows that, although rarely used, it is reliable for the determination of the reduction potential of complex mixtures compared to methods that use unstable free radicals such as DPPH.
  • For the modification, we used a waste product left over from the supercritical extraction with carbon dioxide of hop cones, which can 'refine' the magnetic nanoparticles, which are known 'per se' to have a tendency to aggregate.
  1. UV-Vis spectroscopic data is useless in the presented version.

Answer: We inspired by the following manuscript. The authors obtained similar spectra for IONPs. So, we added the following references. In addition, we added all the spectra of the starting reactants.

Wu, W.; Jiang, C.Z.; Roy, V.A. Designed synthesis and surface engineering strategies of magnetic iron oxide nanoparticles for biomedical applications. Nanoscale 2016, 8, 19421-19474. doi: 10.1039/c6nr07542h.

Madivoli, E.S.; Kareru, P.G.; Maina, E.G.; Nyabola, A.O.; Wanakai, S.I.; Nyang’au, J.O. Biosynthesis of iron nanoparticles using Ageratum conyzoides extracts, their antimicrobial and photocatalytic activity. SN Appl. Sci. 2019, 1, 500. doi:10.1007/s42452-019-0511-7.

Ali, A.; Zafar, H.; Zia, M.; Ul Haq, I.; Phull, A.R.; Ali, J.S.; Hussain, A. Synthesis, characterization, applications, and challenges of iron oxide nanoparticles. Nanotechnol. Sci. Appl. 2016, 9, 49-67. doi: 10.2147/NSA.S99986.

Gobi, M.; Sujatha, M.; Pradeepa, V.; Muralidharan, M.; Venkatesan, M. Green synthesis of iron oxide nanoparticles (FeONPs) and its antibacterial effect using Chamaecrista nigricans (Vahl) Greene (Caesalpiniaceae). Biomass Conv. Bioref. 2023. doi:10.1007/s13399-023-05184-8.

Nadagouda, M.N.; Castle, A.B.; Murdock, R.C.; Hussainb, S.M.; Rajender S. Varma In vitro biocompatibility of nanoscale zerovalent iron particles (NZVI) synthesized using tea polyphenols. Green Chem. 2010, 12, 114–122. doi:10.1039/B921203P.

  1. There is no proof for the formation of IO NP.

Answer: As we mentioned above we are preparing the manuscript basing on our patent with other techniques characterizing the NPs, like XPS. We prepared NPs according to procedure which has been already described in literature. 

  1. SEM images cannot prove the size. DLS may help.

Answer: Thank you for point it out. Size measurements were added by DLS. However, hydrodynamic diameter obtained using DLS technique is not, in our opinion, representative taking into account aggregation of magnetic NPs. However, we agree with the reviewer that this technique is common in papers studied NPs.

  1. HOP extracts have better antibacterial activity than IONP+HOP extracts (especially ethanol extracts), So, it is not useful for further processing.

Answer: We added new data concerning evaluation of activity of IONPs modified using DMSO extract. The obtained results are more promising.  

Round 2

Reviewer 3 Report

Comments and Suggestions for Authors

I would like to request to add DLS figure. It is necessary to fix the abbreviations and units.

Comments on the Quality of English Language

Minor editing is required.

Author Response

Reviewer: I would like to request to add DLS figure. It is necessary to fix the abbreviations and units.

Answer: Thank You very much for this suggestion. It was done.